# A Novel Lattice Boltzmann Scheme with Single Extended Force Term for Electromagnetic Wave Propagating in One-Dimensional Plasma Medium

**Huifang Ma [1], Bin Wu [2], Ying Wang [1], Hao Ren [1], Wanshun Jiang [2,\*], Mingming Tang [3,\*] and Wenyue Guo [1,\*]**

[1]  School of Materials Science and Engineering, China University of Petroleum (East China),
    Qingdao 266580, China; mahuif@126.com (H.M.); wangying251x@163.com (Y.W.); renh@upc.edu.cn (H.R.)
[2]  The 41st Institute of China Electronics Technology Group Corporation, Qingdao 266555, China;
    wubinw@126.com
[3]  School of Geosciences, China University of Petroleum (East China), Qingdao 266580, China
\*  Correspondence: jiangwanshun@ceyear.com (W.J.); tangmingming@upc.edu.cn (M.T.);
    wyguo@upc.edu.cn (W.G.); Tel.: +86-0532-86880796 (W.J. & W.G.); +86-131-56891161 (M.T.)

**Abstract:** A one-dimensional plasma medium is playing a crucial role in modern sensing device design, which can benefit significantly from numerical electromagnetic wave simulation. In this study, we introduce a novel lattice Boltzmann scheme with a single extended force term for electromagnetic wave propagation in a one-dimensional plasma medium. This method is developed by reconstructing the solution to the macroscopic Maxwell's equations recovered from the lattice Boltzmann equation. The final formulation of the lattice Boltzmann scheme involves only the equilibrium and one non-equilibrium force term. Among them, the former is calculated from the macroscopic electromagnetic variables, and the latter is evaluated from the dispersive effect. Thus, the proposed lattice Boltzmann scheme directly tracks the evolution of macroscopic electromagnetic variables, which yields lower memory costs and facilitates the implementation of physical boundary conditions. Detailed conduction is carried out based on the Chapman–Enskog expansion technique to prove the mathematical consistency between the proposed lattice Boltzmann scheme and Maxwell's equations. Based on the proposed method, we present electromagnetic pulse propagating behaviors in nondispersive media and the response of a one-dimensional plasma slab to incident electromagnetic waves that span regions above and below the plasma frequency $\omega_p$, and further investigate the optical properties of a one-dimensional plasma photonic crystal with periodic thin layers of plasma with different layer thicknesses to verify the stability, accuracy, and flexibility of the proposed method.

**Keywords:** lattice Boltzmann scheme; plasma medium; one-dimensional plasma photonic crystal; electromagnetic wave

## 1. Introduction

Recently, plasma media have been widely used in current photonic topological insulator studies [1–3], particularly in one-dimensional (1D) plasma photonic crystals (PhCs) [4–7], two-dimensional (2D) PhCs [8,9], and three-dimensional (3D) PhC construction. Nevertheless, owing to their complex design and manufacturing, the potential applications of topological photonics in 2D and 3D PhCs are very limited. Consequently, 1D PhCs are preferred because of their great maneuverability and ease of manufacture [10,11]. 1D plasma PhCs are used to realize the surface impedance and bulk band. Furthermore, 1D plasma PhCs have been employed to manipulate the light-matter interactions by tuning the plasma layer thickness in PhCs.

Owing to the advantages of 1D plasma medium applications, researchers are strongly motivated to investigate the electromagnetic response on plasma dispersive media to design more efficient 1D PhCs sensing devices. In recent decades, several numerical methods

have been proposed to solve the electromagnetic (EM) wave propagation problem [12–14]. Among these, the finite-difference time-domain (FDTD) method has been proven to be one of the most effective approaches [14]. This method, first proposed by Yee [15], is mainly based on the finite-difference scheme to discretize time-dependent Maxwell's equations. This method works well in the media of linear, homogeneous, isotropic, and non-dispersive materials, but is not suitable for the simulation of EM wave in frequency-dependent dispersive media [16,17]. In addition to organic matter, the more common water film is also a dispersive material. Alternative methods have been introduced for EM waves in dispersive media, for example, recursive convolution [17,18] and the auxiliary differential equation method [19,20]. The FDTD method proposed by Luebbers and Kunz is the most promising method for solving dispersive media problems [21,22].

In the last two decades, the lattice Boltzmann method (LBM) has been used as an alternative for the simulation of partial differential equations (PDEs) [23]. It has been widely used for various types of linear and non-linear PDEs, such as diffusion [24], flow [25], waves [26], quantum mechanics [27–29] and heat transfer [30,31]. Moreover, with advanced LBM technique, such as the immersed boundary-LBM method, LBM has also been used to determine hydrodynamic force and energy exchange problems among particle and flow [32,33]. Given the paramount role of EM phenomena in science and technology, it is of great interest to investigate whether the LBM is able to improve simulations of complex EM phenomena. LBMs have been used to simulate EM propagation in non-dispersive media based on the scalar distribution function [29,34] and vector distribution function [35,36]. However, to the best of our knowledge, few approaches have been implemented to solve the EM problem in dispersive media with LBM. Chen et al. [16] developed a lattice model with pseudo permittivity and two force terms for terahertz electromagnetic waves propagation in one-dimensional dispersive media, but the accuracy of the pseudo permittivity in their model is related to the time step value, and each of the two force terms needs to be calculated based on a two-step iteration algorithm.

In this study, we introduce a different expression form for the Maxwell's equation in an isotropic dispersive medium and propose a new frequency-dependent LBM with only a single extended force term (LBM-SEF) for electromagnetic waves propagation in a plasma medium. Thereafter, two numerical validation models are executed by comparing the analytical and existing FDTD solutions. Furthermore, we studied the ability of the proposed lattice Boltzmann scheme to generate electromagnetic waves propagation in a 1D PhC.

## 2. Materials and Methods

The LBM is a general method for describing the evolution of the particle distribution in discrete space and time [23]. The density and moments calculated based on particle distributions can precisely represent the density and velocity to be modeled [24,25]. In this study, an extended LBM equation with a forcing term for capturing conductivity characteristics is proposed. Thereafter, the consistency between the Maxwell and the LBM equations is demonstrated through the Chapman–Enskog multi-scale expansion technique.

### 2.1. Theoretical Model of Plasma Media

To describe the dispersive behavior of the plasma medium, we start by describing the motion of a free electron in such media:

$$m\frac{d^2x}{dt^2} + m\vartheta_c\frac{dx}{dt} = -eE_0\exp(i\omega t) \tag{1}$$

where $x$ is the displacement of the electron, $m$ is the electron mass, $\vartheta_c$ is the damping frequency, $e$ is the charge of the electron, $E_0$ is the amplitude of the external electric field, and $\omega$ is the angular frequency of the external electric field. By solving Equation (1), we obtain the Drude model of the plasma media as:

$$\varepsilon(\omega) = 1 - \frac{\omega_p^2}{\omega(\omega - i\vartheta_c)} \tag{2}$$

where $\omega_p$ stands for the plasma frequency.

### 2.2. Governing Equations

The governing time-dependent Maxwell's equations for dielectric materials at a particular location x and time t are as follows [29]:

$$\nabla \times \mathbf{E}(x,t) = -\frac{\partial \mathbf{B}}{\partial t}, \ \nabla \times \mathbf{H}(x,t) = -\frac{\partial \mathbf{D}(x,t)}{\partial t} \tag{3}$$

where $\mathbf{H}$ and $\mathbf{E}$ are the magnetic field intensity and electric field intensity, respectively, and $\mathbf{B}$ and $\mathbf{D}$ represent the magnetic induction and electric displacement, respectively. The relationship between $\mathbf{B}$ and $\mathbf{H}$, $\mathbf{D}$, and $\mathbf{E}$ is given by $\mathbf{B} = \mu\mathbf{H}$, $\mathbf{D} = \varepsilon\mathbf{E}$, where $\mu$ and $\varepsilon$ are the permittivity and permeability of the media, respectively. $\mu$ and $\varepsilon$ are defined with relative constants $\mu_r$, $\varepsilon_r$ as $\mu = \mu_r\mu_0$, $\varepsilon = \varepsilon_r\varepsilon_0$.

For a linear isotropic dispersive medium, the dielectric constant is expressed as $\varepsilon(\omega)$, and the relationship between $\mathbf{D}$ and $\mathbf{E}$ in the frequency domain is expressed as $\mathbf{D}(\omega) = \varepsilon(\omega)\mathbf{E}(\omega)$. The relationship in the time domain of $\mathbf{D}(x,t)$ and $\mathbf{E}(x,t)$ could be written as

$$\mathbf{D}(x,t) = \varepsilon_\infty\varepsilon_0\mathbf{E}(x,t) + \varepsilon_0 \int_0^t \mathbf{E}(x,t-\tau)\zeta(\tau)\mathrm{d}\tau \tag{4}$$

where $\varepsilon_\infty$ is the infinite-frequency permittivity, and $\zeta(\tau)$ is the electric susceptibility in the time domain. Compared with Chen et al. [16], we use a different expression form for the Maxwell's equation based on Equation (3):

$$\varepsilon_\infty\varepsilon_0\frac{\partial \mathbf{E}(x,t)}{\partial t} = \nabla \times \mathbf{H}(x,t) - \varepsilon_0\frac{\partial}{\partial t} \int_0^t \mathbf{E}(x,t-\tau)\zeta(\tau)\mathrm{d}\tau$$

$$\frac{\partial \mathbf{B}}{\partial t} = -\nabla \times \mathbf{E}(x,t) \tag{5}$$

In the studies by Chen et al. [16], it is necessary to calculate the pseudo permittivity in advance before simulating the terahertz electromagnetic wave propagation, while the accuracy of the pseudo permittivity is related to the simulation time step $\Delta t$ used in special cases [16]. Based on Equation (5), it is no longer necessary to use pseudo permittivity. It can be observed from the following mathematical derivation process that only one force term is needed, without the use of pseudo permittivity.

For 1D dispersive media, we can obtain an additional simplified model for planar electromagnetic wave propagation:

$$\varepsilon_\infty\varepsilon_0\frac{\partial E_y(x,t)}{\partial t} = \frac{\partial H_z(x,t)}{\partial x} - \varepsilon_0\frac{\partial}{\partial t} \int_0^t E_y(x,t-\tau)\zeta(\tau)\mathrm{d}\tau$$

$$\mu\frac{\partial H_z(x,t)}{\partial t} = -\frac{\partial E_y(x,t)}{\partial x} \tag{6}$$

It is worth noting that with Equation (6), we obtain not only one stricter model, but also a more simplified LBM model, which will be discussed in the following section.

### 2.3. The Extended LBM

Herein, we propose a dimensionless LBM involving a new special force term for linear dispersive media. The proposed LBM is expressed as:

$$f_i(x + \boldsymbol{e_i}\Delta t, t + \Delta t) - f_i(x,t) = -\frac{\Delta t}{\tau}\left(f_i(x,t) - f_i^{eq}(x,t)\right) - \Delta t F_i(x,t) \ i = 0,1,2,\ \ldots,\ b \tag{7}$$

where $f_i(x,t)$ is the particle distribution function in direction $i$, location $x$, and time $t$. $\mathbf{e_i}$ is the lattice velocity vector in the $i$-th direction, $\Delta t$ is the lattice time step, and $\tau$ is the relaxation time. EM wave equation has no dissipation term, i.e., kinematic viscosity is zero, this is achieved by setting the relaxation time $\tau = 1/2$, which is same with the proposed value in the studies by Chen et al. [16] and Dhuri and Hanasoge (2017) [34]. b is the lattice vector number, which is 2 for 1D media. $f_i^{eq}(x,t)$ is the local equilibrium distribution. $F_i(x,t)$ is the dispersive forcing term proposed in this study.

The macroscopic quantities in Equation (6) are defined by:

$$\varepsilon_\infty \varepsilon_0 E_y(x,t) = \sum_i f_i(x,t) \quad H_z(x,t) = \sum_i e_i f_i(x,t) \tag{8}$$

To solve for the EM wave, we refer to the LBM model for calculating the bandgap of photonic materials and $f_i^{eq}(x,t)$ is written as

$$f_i^{eq}(x,t) = A_i \varepsilon_\infty \varepsilon_0 E_y(x,t) + B_i e_i H_z(x,t) \tag{9}$$

where $A_i$ and $B_i$ are distribution weights. Considering the 1D physical symmetry, we have

$$A_i = A, B_i = B, \; i > 0 \tag{10}$$

and the equilibrium distribution $f_i^{eq}(x,t)$ is chosen as

$$f_i^{eq}(x,t) = A_0 \varepsilon_\infty \varepsilon_0 E_y(x,t), \quad i = 0$$

$$f_i^{eq}(x,t) = A \varepsilon_\infty \varepsilon_0 E_y(x,t) + B e_i H_z(x,t), \quad i > 0 \tag{11}$$

Based on Equations (8) and (10), the following equation is obtained:

$$\sum_i A_i = 1 \quad \sum_{i>0} B_i = 1 \tag{12}$$

Next, we used the Chapman–Enskog expansion method to determine distribution weights.

*2.4. Chapman–Enskog Expansion*

Based on the Chapman–Enskog expansion technique, the particle distribution $f_i(x,t)$ can be expanded up to the second order with the expansion parameter θ:

$$f_i(x,t) = f_i^{eq}(x,t) + \theta f_i^1(x,t) + \theta^2 f_i^2(x,t) + O(\theta^3) \tag{13}$$

It is worth noting that in Equation (10), $f_i^1(x,t)$ and $f_i^2(x,t)$ could be treated as formal expansion functions representing the distribution function at different scales. We do not need the exact formula of these functions when using the Chapman–Enskog expansion technique.

Using Equations (11) and (13), we obtain the following equations:

$$\sum_i f_i(x,t) = \sum_i f_i^0(x,t) + \theta f_i^1(x,t) + \theta^2 f_i^2(x,t) + O(\theta^3) = \sum_i f_i^{eq}(x,t)$$

$$\sum_i e_i f_i(x,t) = \sum_i e_i f_i^0(x,t) + \theta e_i f_i^1(x,t) + \theta^2 e_i f_i^2(x,t) + e_i O(\theta^3) = \sum_i \mathbf{e_i} f_i^{eq}(x,t) \tag{14}$$

where $f_i^0(x,t) = f_i^{eq}(x,t)$. Since $e_0$ is a zero vector and θ is an arbitrarily small value, we can obtain the following:

$$\sum_i f_i^k(x,t) = 0$$

$$\sum_i e_i f_i^k(x,t) = 0, \text{ for } k > 0 \tag{15}$$

Using the Taylor expansion technique, we obtain the following equation based on the left-hand side of Equation (7):

$$
\begin{aligned}
&\mathrm{f}_i(x + e_i\Delta t, t + \Delta t) - \mathrm{f}_i(x, t) \\
&= \Delta t(\partial_t + e_{i\alpha}\partial_{x\alpha})\mathrm{f}_i(x,t) + \tfrac{\Delta t^2}{2}(\partial_t + e_{i\alpha}\partial_{x\alpha})^2\mathrm{f}_i(x,t) + O(\Delta t^3)
\end{aligned}
\tag{16}
$$

While the forcing term in Equation (7) could also be written as:

$$
\mathrm{F}_i(x,t) = \frac{\partial}{\partial t}\int_{t=0}^{t}\mathrm{F}_i(x,\tau)d\tau = \frac{\partial}{\partial t}\widetilde{\mathrm{F}}_i(x,t)
\tag{17}
$$

Using the Chapman–Enskog expansion technique, the scales of the derivations in time $\partial_t$ and space $\partial_{x\alpha}$ in Equation (16) can be written as

$$
\partial_t = \theta\partial_{t(0)} + \theta^2\partial_{t(1)} + O(\theta^3)
$$

$$
\partial_{x\alpha} = \theta\partial_{x\alpha(1)} + O(\theta^2)
\tag{18}
$$

While the forcing term is assumed as:

$$
\widetilde{\mathrm{F}}_i(x,\tau) = \theta\widetilde{\mathrm{F}_{i(1)}}(x,\tau) + O(\theta^2)
\tag{19}
$$

Using the aforementioned scales and substituting Equations (16) and (18) into Equation (7), we obtain:

$$
\begin{aligned}
&\Delta t(\partial_t + e_{i\alpha}\partial_{x\alpha})\mathrm{f}_i(x,t) + \tfrac{\Delta t^2}{2}(\partial_t + e_{i\alpha}\partial_{x\alpha})^2\mathrm{f}_i(x,t) + O(\Delta t^3) \\
&= -\tfrac{\Delta t}{\tau}\left(\mathrm{f}_i(x,t) - \mathrm{f}_i^0(x,t)\right) - \Delta t\mathrm{F}_i(x,t) \\
&= -\tfrac{\Delta t}{\tau}\left(\theta\mathrm{f}_i^1(x,t) + \theta^2\mathrm{f}_i^2(x,t) + O(\theta^3)\right) - \Delta t\left(\theta\partial_{t(0)} + \theta^2\partial_{t(1)} + O(\theta^3)\right)\left(\theta\widetilde{\mathrm{F}_{i(1)}}(x,t) + O(\theta^2)\right) \\
&= -\tfrac{\Delta t}{\tau}\left(\left(\mathrm{f}_i^0(x,t) + \theta\mathrm{f}_i^1(x,t) + \theta^2\mathrm{f}_i^2(x,t) + O(\theta^3)\right) - \mathrm{f}_i^0(x,t)\right) - \Delta t\theta\mathrm{F}_{i(1)}(x,t)
\end{aligned}
\tag{20}
$$

Grouping terms based on Chapman–Enskog order $\theta$ and $\theta^2$, leading to:

$$
\partial_{t(0)}\mathrm{f}_i^0(x,t) + \mathbf{e_{i\alpha}}\partial_{x\alpha(1)}\mathrm{f}_i^0(x,t) = -\frac{1}{\tau}\mathrm{f}_i^1(x,t)
$$

$$
\begin{aligned}
&\partial_{t(1)}\mathrm{f}_i^0(x,t) + \left(-\tau + \tfrac{\Delta t}{2}\right)(\partial_{t(0)} + e_{i\alpha}\partial_{x\alpha(1)})^2\mathrm{f}_i^0(x,t) \\
&= -\frac{1}{\tau}\mathrm{f}_i^2(x,t) - \partial_{t(0)}\widetilde{\mathrm{F}_{i(1)}}(x,t)
\end{aligned}
\tag{21}
$$

Summing Equation (21) over $i$ and using Equations (11) and (15), we obtain:

$$
\begin{aligned}
&\partial_{t(0)}\sum_i\mathrm{f}_i^0(x,t) + \partial_{x\alpha(1)}\sum_i e_{i\alpha}\mathrm{f}_i^0(x,t) = -\frac{1}{\tau}\sum_i\mathrm{f}_i^1(x,t) \\
&= \partial_{t(0)}\varepsilon_\infty\varepsilon_0\mathrm{E}_y(x,t) + \partial_{x\alpha(1)}\mathrm{H}_z(x,t) = 0
\end{aligned}
\tag{22}
$$

and

$$
\begin{aligned}
&\partial_{t(1)}\varepsilon_\infty\varepsilon_0\mathbf{E_y}(x,t) + \left(-\tau + \tfrac{\Delta t}{2}\right)\left(\begin{array}{c}\partial_{t(0)}\partial_{t(0)}\varepsilon_\infty\varepsilon_0\mathrm{E}_y(x,t) \\ +2\partial_{t(0)}\partial_{x\alpha(1)}\mathrm{H}_z(x,t) + \partial_{x\alpha(1)}\partial_{x\alpha(1)}2A\varepsilon_\infty\varepsilon_0\mathrm{E}_y(x,t)\end{array}\right) \\
&= -\partial_{t(0)}\sum_i\widetilde{\mathrm{F}_{i(1)}}(x,t)
\end{aligned}
\tag{23}
$$

Multiplying Equations (22) and (23) by $e_{i\alpha}$, and considering (15), we have

$$
\partial_{t(0)}\sum_i e_{i\alpha}\mathrm{f}_i^0(x,t) + \partial_{x\alpha(1)}\sum_i e_{i\alpha}e_{i\alpha}\mathrm{f}_i^0(x,t) = -\frac{1}{\tau}\sum_i e_{i\alpha}\mathrm{f}_i^1(x,t)
$$

$$
\partial_{t(0)}\mathrm{H}_z(x,t) + \partial_{x\alpha(1)}2A\varepsilon_\infty\varepsilon_0\mathrm{E}_y(x,t) = 0
\tag{24}
$$

and

$$\partial_{t^{(1)}} H_z(x,t) + \left(-\tau + \tfrac{\Delta t}{2}\right)\left(H_z(x,t) + 4\partial_{t^{(0)}}\partial_{x\alpha^{(1)}}\varepsilon_\infty\varepsilon_0 E_y(x,t) + \partial_{x\alpha^{(1)}}\partial_{x\alpha^{(1)}} H_z(x,t)\right)$$
$$= -\partial_{t^{(0)}} \sum_i e_{i\alpha}\widetilde{F_{i^{(1)}}}(x,t) \tag{25}$$

By comparing with Maxwell's Equation (6), we can use the following constraints:

$$\sum_i \widetilde{F_{i^{(1)}}}(x,t) = \varepsilon_0 \int_0^t E_y(x,t-\tau)\zeta(\tau)\mathrm{d}\tau$$

$$2A\varepsilon_\infty\varepsilon_0 = \frac{1}{\mu}$$

$$\tau = \frac{\Delta t}{2}$$

$$\sum_i e_{i\alpha}\widetilde{F_{i^{(1)}}}(x,t) = 0 \tag{26}$$

and the coefficients in Equation (10) are determined as:

$$A = \frac{1}{2\varepsilon_\infty\varepsilon_0\mu}$$

$$A_0 = 1 - 2A = \frac{\varepsilon_\infty\varepsilon_0\mu - 1}{\varepsilon_\infty\varepsilon_0\mu}$$

$$B = \frac{1}{2} \tag{27}$$

By taking ($\theta \times$ Equation (22) + $\theta^2 \times$ Equation (23)), we have:

$$\varepsilon_\infty\varepsilon_0\partial_t E_y(x,t) = -\partial_x H_z(x,t) - \varepsilon_0\partial_t \int_0^t E_y(x,t-\tau)\zeta(\tau)\mathrm{d}\tau + O(\Delta t^3)$$
$$+ O(\theta^3)$$

$$\mu\partial_t H_z(x,t) = -\partial_x E_y(x,t) + O(\Delta t^3) + O(\theta^3) \tag{28}$$

From Equation (28), it can be proven that with the newly proposed lattice Boltzmann forcing term, we can recover the Maxwell's equations as $\Delta t$ and $\theta$ approach zero.

The introduced integrate forcing term $\widetilde{F_{i^{(1)}}}(x,t)$ is determined to be:

$$\widetilde{F_1}(x,t) = \widetilde{F_2}(x,t) = 0$$

$$\widetilde{F_0}(x,t) = \varepsilon_0 \int_0^t E_y(x,t-\tau)\zeta(\tau)\mathrm{d}\tau \tag{29}$$

Based on Equation (29), we can calculate the forcing term:

$$F_1(x,t) = F_2(x,t) = 0$$

$$F_0(x,t) = \varepsilon_0\partial_t \int_0^t E_y(x,t-\tau)\zeta(\tau)\mathrm{d}\tau$$
$$= \varepsilon_0 E_y(x,0)\zeta(t) + \varepsilon_0 \int_0^t \partial_t E_y(x,t-\tau)\zeta(\tau)\mathrm{d}\tau \tag{30}$$

We refer to the Drude model used in the time domain (Chen et al., 2013) [16]:

$$\zeta(t) = \frac{\omega_p^2}{\vartheta_c}(1 - \exp(-\vartheta_c t))U(t) \tag{31}$$

where $\vartheta_c$ is the damping frequency, and $\omega_p$ stands for the plasma frequency.

Based on Equations (30) and (31), the forcing term is determined by $P(x, t)$:

$$
\begin{aligned}
P(x,t) &= \int_0^t \partial_t E_y(x, t - \tau)\zeta(\tau)\mathrm{d}\tau = \int_0^{n\Delta t} \partial_t E_y(x, n\Delta t - \tau)\zeta(\tau)\mathrm{d}\tau \\
&= \sum_{m=0}^{n} \partial_t E_y(x, (n-m)\Delta t) \int_{m\Delta t}^{(m+1)\Delta t} \zeta(\tau)\mathrm{d}\tau \\
&= \sum_{m=0}^{n} \partial_t E_y(x, (n-m)\Delta t) \left[\frac{\omega_p^2}{\vartheta_c}\tau + \frac{\omega_p^2}{\vartheta_c^2}\exp(-\vartheta_c\tau)\right]\Big|_{m\Delta t}^{(m+1)\Delta t} \\
&= \sum_{m=0}^{n} \partial_t E_y(x, (n-m)\Delta t) \left[\frac{\omega_p^2}{\vartheta_c}\Delta t + \frac{\omega_p^2}{\vartheta_c^2}\exp(-\vartheta_c m\Delta t)(\exp(-\vartheta_c\Delta t) - 1)\right]
\end{aligned}
\tag{32}
$$

The above method is constructed for 1D EM wave simulation problem, where we do not need to consider the anisotropic heterogeneity problem. However, when it is necessary to solve 2D or 3D problems, the method proposed in this paper cannot be directly applied to anisotropic heterogeneity problems, but we can use permeability tensors $\varepsilon(\omega)$ of 2D or 3D anisotropic material and calculated the force terms for 2D and 3D anisotropic dispersion materials by following Equations (4) and (29)–(32). In this study, we focus on 1D EM wave simulation problems.

A new issue in this study that one must perform conversion between LBM and physical spaces. Using light speed as a critical unit conversion parameter, Table 1 summaries the conversion rules between LBM quantities (superscript "*LB*") and their corresponding physical values (superscript "*py*").

**Table 1.** Conversion between LBM and physical space.

| Denomination | LBM Context | Physical Context |
|:---:|:---:|:---:|
| Space step | $\Delta x^{LB} = 1$ | $\Delta x^{py} = \Delta x^{LB}\frac{DL}{L}$ |
| Time step | $\Delta t^{LB} = 1$ | $\Delta t^{py} = \Delta t^{LB}\frac{\Delta x^{py}}{\Delta x^{LB}}\frac{c^{LB}}{c^{py}}$ |
| Light speed | $c^{LB} = 1$ | $c^{py} = c$ |
| Electric field density | $E^{LB} = 1$ | $E^{py} = E^{LB}\theta^V$ |
| Frequency | $f^{LB} = 1$ | $f^{py} = f^{LB}\frac{\Delta t^{LB}}{\Delta t^{py}}$ |

($c$ is the light speed, $DL$ is the domain length, and $L$ is the cell number, $\theta^V$ = 1 kV/m).

## 3. Results

The accuracy of the proposed LBM is validated and demonstrated using a new forcing term. Three typical cases, with available analytical or numerical solutions, are considered. The simulations showed that the proposed model reproduced the correct electromagnetic propagation in non-dispersive and dispersive media.

### 3.1. Electromagnetic Pulse in Non-Dispersive Media

As the first benchmark, we simulate the propagation of a terahertz electromagnetic Gaussian pulse crossing a dielectric interface in a 1D array of L cells with periodic boundary conditions. One half of the simulation space, $x < L/2$ is a vacuum ($\varepsilon = \varepsilon_0$), and the other half, $x > L/2$, represents a non-dispersive medium with a dielectric constant of $\varepsilon_r = 2.0$. Referring to previous studies [16,36], the pulse is described by

$$
E(x,t) = E_M \exp\left(-[(x - x_c)/\alpha]^2\right), \quad H(x,t) = H_M \exp\left(-[(x - x_c)/\alpha]^2\right)
\tag{33}
$$

where the constant $\alpha$ fixes the pulse width, and $E_M$ and $H_M$ are the pulse amplitudes of the electric and magnetic field intensities, respectively. We chose L = 800, c = 1, $E_M$ = 1000, $\alpha = 30$, and $x_c = 250$. The initial conditions and electric field after 300 cycles are shown in Figure 1.

The theoretical predictions for the amplitude of the transmitted $E'_M$ and reflected pulses $E''_M$ can be computed from the incident pulse $E_M^0$ [36,37]:

$$\frac{E'_M}{E^0_M} = \frac{2}{\sqrt{\frac{\varepsilon'_r}{\varepsilon^0_r}} + 1}, \quad \frac{E''_M}{E^0_M} = \frac{\sqrt{\frac{\varepsilon'_r}{\varepsilon^0_r}} - 1}{\sqrt{\frac{\varepsilon'_r}{\varepsilon^0_r}} + 1} \tag{34}$$

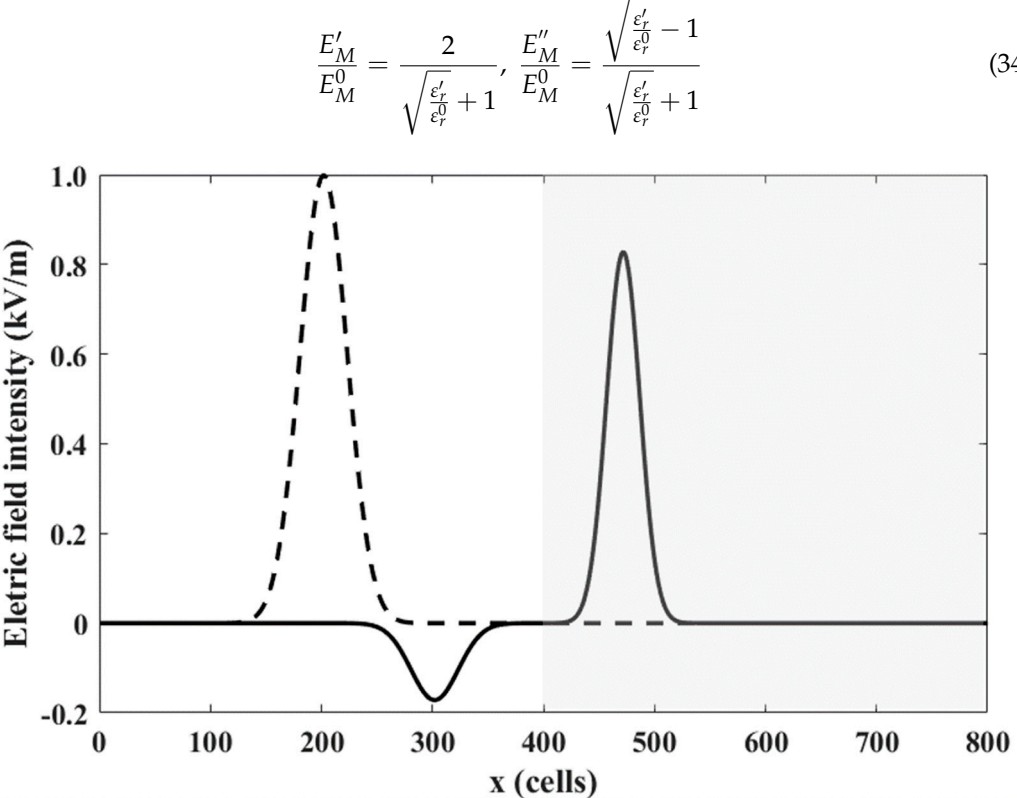

**Figure 1.** Distribution of electric pulse crossing a dielectric interface. The shadow zone is the dielectric media, with dielectric constant $\varepsilon_r = 2.0$ and the other one corresponds to the media with $\varepsilon_r = 1.0$. The curves are the intensity of the electric field at t = 0 (dashed line), and t = 300 (solid line).

Based on Equation (32), the amplitude ratio of the transmitted pulse to the incident pulse is theoretically $\frac{E'_M}{E^0_M} = 0.82843$, whereas that for the reflected pulse to the incident pulse should be $\frac{E''_M}{E^0_M} = 0.17157$. Based on the proposed model, the computed values for these two ratios were 0.82761 and 0.17153, respectively. Thus, the computed values agreed well with the theoretical values, and the relative errors were less than 1%.

Inspired by the method proposed by Dhuri et al. (2017) [38], we further analyze the dispersion characteristics of the proposed 1D LBM method by studying the response of the scheme to plane waves in homogeneous non-dispersion media. Figure 2 compares the dispersion relations in the FDTD [39] and proposed LBM method with the exact dispersion relations for a one-dimensional problem. In these figures, the normalized wave number $K = 2k\Delta x$ is shown as a function of the normalized frequency $W = \omega \Delta t$.

Figure 2 shows that the dispersion relation of proposed LBM method in this study agrees very well with the theoretical exact solution curve. Furthermore, Figure 2 also implies that the proposed LBM method has smaller dispersion error than FDTD method under very high wave number condition ($K > 2.0$), which is another potential advantage of the LBM over FDTD method for the particular problem discussed in this study. However, the 1D LBM method proposed in this study used three particle distribution functions ($f_0(x,t), f_1(x,t), f_2(x,t)$) to represent two physical fields (H and E); thus, compared with the 1D FDTD method, the LBM method uses one-third more memory resources.

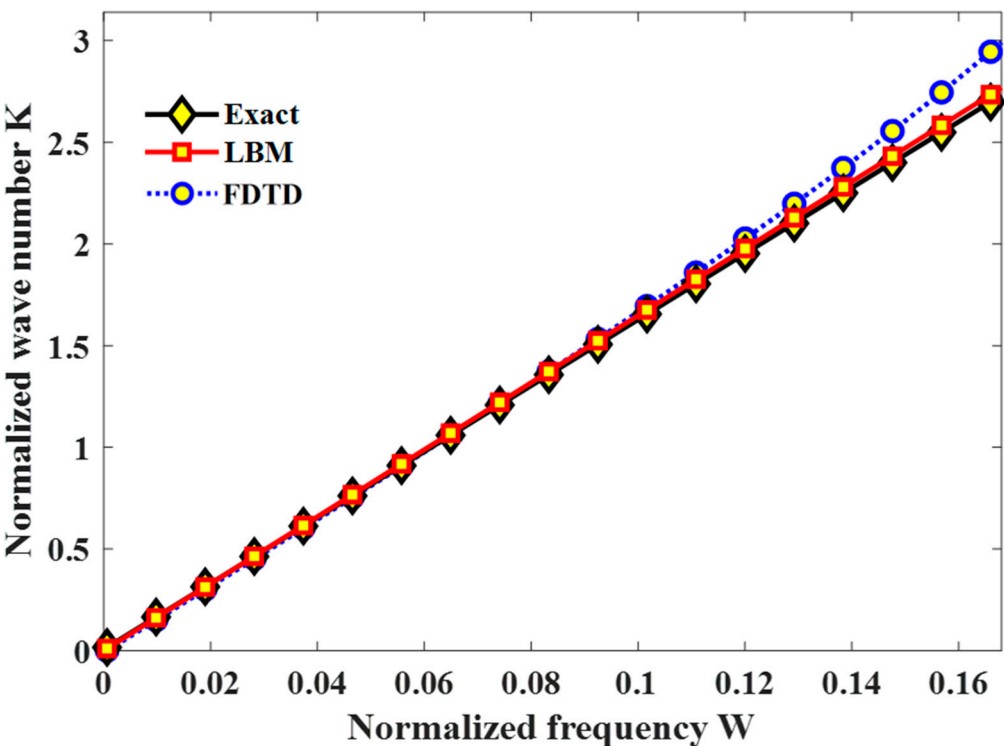

**Figure 2.** Comparison of dispersion relations in the FDTD and proposed LBM method in this study with the exact dispersion relation in non-dispersive media.

### 3.2. Effects of Plasma Frequency on EM Waves

In this study, a pulse propagating in free space upon the plasma was considered. The properties of plasma from silver are used in Sections 3.2 and 3.3, with a plasma frequency $\omega_p = 2000$ THz and a damping frequency $\vartheta_c = 50$ THz. Figure 3 shows that at a low frequency $\frac{\omega}{2\pi} = 500$ THz, the permittivity of the plasma medium is in the negative permittivity region, and at a high frequency $\frac{\omega}{2\pi} = 4000$ THz, the plasma medium's permittivity is close to that of vacuum.

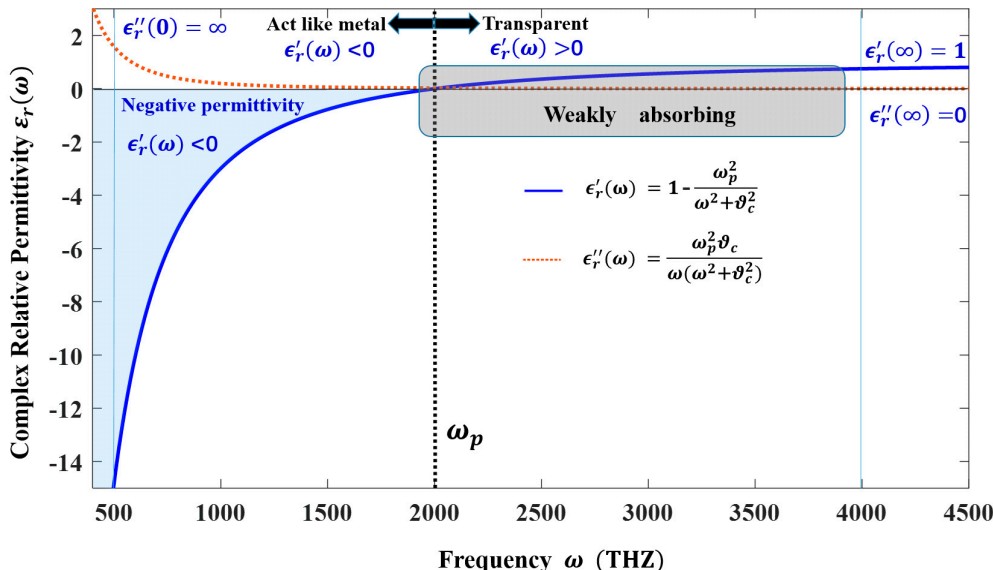

**Figure 3.** Complex relative permittivity of plasma in frequency domain.

We further investigated the phenomenon of plasma frequency of the plasma medium to an incoming electromagnetic wave to demonstrate the adaptability of the proposed LBM-SEF method. A pulse propagating in air that came upon the plasma medium was considered in this study. As shown in Figure 4, the entire simulation domain, with a length of 10 μm, is evenly divided into 10,000 lattice cells, and the size of the lattice is uniform, as each had a length of 1 nm. The EM pulse in the air domain is assumed to be a Gaussian-modulated sinusoidal pulse with amplitude expressed as follows:

$$\mathrm{E}(x,t) = E_M \sin(2\pi f_c t) \exp\left(-[(x - x_c)/\alpha]^2\right) \tag{35}$$

where the constant $\alpha$ fixes the pulse width, $E_M$ is the pulse amplitude of the electric field intensity and magnetic field intensity, $f_c$ is the frequency of the sine function in factor which is used to modulate the center frequency of the EM pulse. $f_c$ is taken to be either 500 THz for the low-frequency case or 4000 THz for the high-frequency case. The incident wave propagates from the right air medium, passes through the middle film of the plasma medium domain, and returns to the air medium again, as shown in Figure 3.

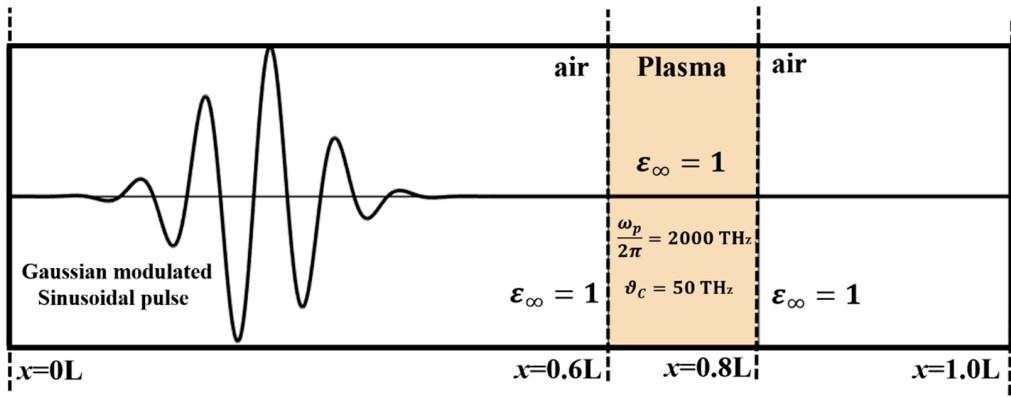

**Figure 4.** Schematic computational domain and incident Gaussian-modulated sinusoidal pulse through a film of the dispersive medium.

We further analyzed the effect of frequency on the EM wave propagation behavior. The simulation results for the low- and high-frequency Gaussian-modulated sinusoidal pulses are presented in Figures 5 and 6. Figure 5a shows that at time $t = 0$ fs, the initial Gaussian-modulated sinusoidal pulse at a frequency of 500 THz is in the air zone. It then starts to touch the air-plasma interface at approximately $t = 11$ fs, as shown in Figure 5b. At $t = 21$ fs, Figure 5c shows that the EM waves are completely reflected back by the plasma slab. When the frequency of the incoming EM waves is 4000 THz, the propagating EM waves in the computational domain are as shown in Figure 6. At time $t = 0$ fs, the initial Gaussian-modulated sinusoidal pulse at a frequency of 500 THz was in the air zone. At time $t = 17$ fs, Figure 6b shows that when the EM waves propagate through the left boundary of the plasma slab, only a very small part of the incoming EM waves is reflected back into the air domain, and most of the other parts penetrate the plasma. Figure 6c shows that at $t = 23$ fs, the EM waves pass through the left boundary of the plasma slab and still move toward the right. Thus, if the frequency of the EM waves is low, the incoming EM waves are completely shielded, and the plasma slab shows the screening effect. When the frequency of the EM waves is sufficiently high, the plasma slab becomes a transparent medium. These simulations further demonstrate that the proposed LBM-SEF method can accurately capture the fundamental characteristics of EM waves in dispersive media.

As we are dealing with conservative equations in differential form, we must calculate the order of convergence of the model. We tracked how the accuracy of a physical variable changed when the resolution of the lattice grid increased, while the time resolution remained the same. The electromagnetic energy density is used to track the changes.

$$E = \frac{1}{2}\left(\varepsilon_r \vec{E}.\vec{E} + \frac{1}{\mu_R}\vec{B}.\vec{B}\right) \tag{36}$$

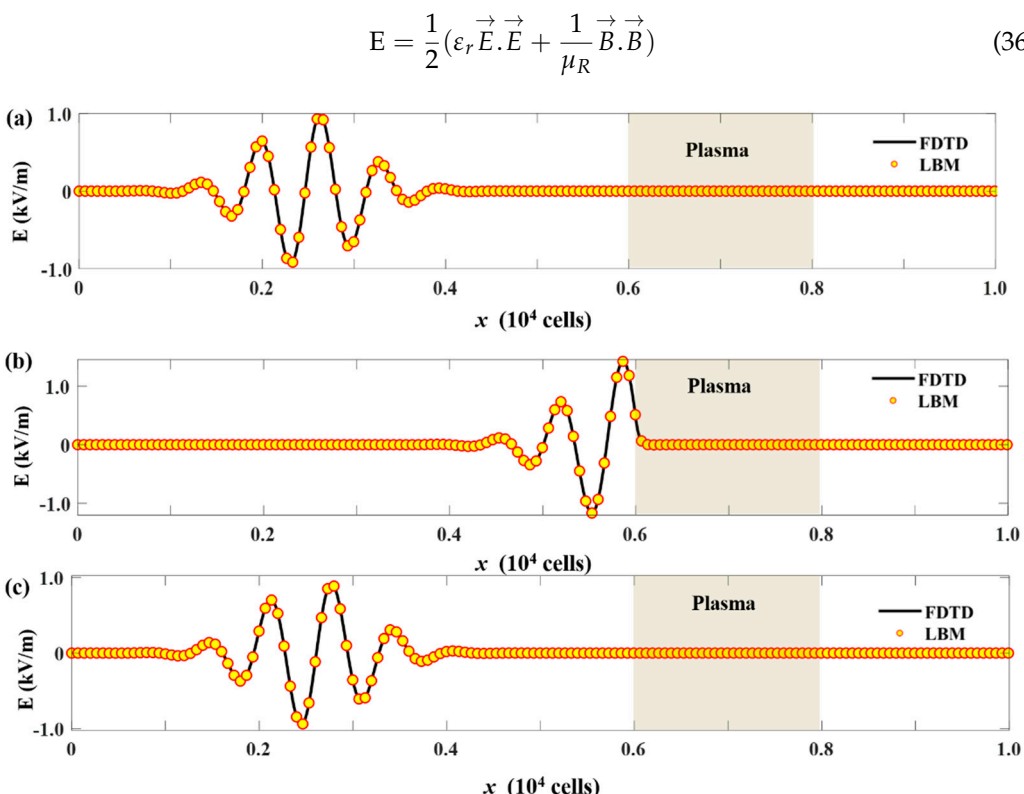

**Figure 5.** Simulation of a wave propagating in air and striking a plasma medium. The plasma has the properties of silver: $\omega_p = 2000$ THz, $\vartheta_c = 50$ THz. The propagating wave has a center frequency of 500 THz. (**a,b**) show electric field in the computational domain at different times (**a**) = 0 fs, (**b**) = 11 fs, and (**c**) = 21 fs. (Left side with film thickness of 1000 cells, right side with film thickness of 2000 cells).

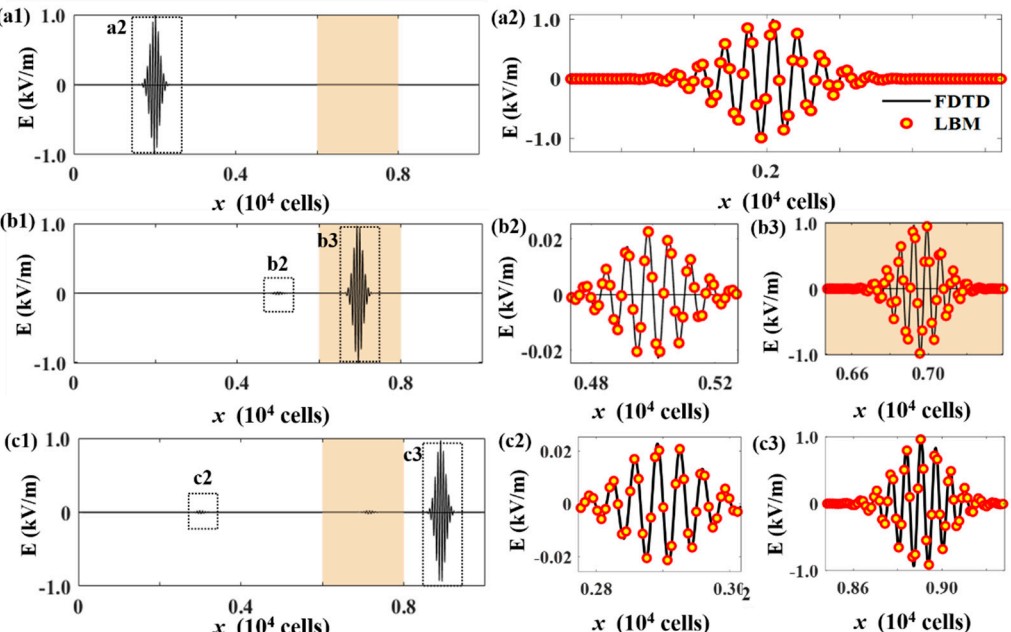

**Figure 6.** Simulation of a wave propagating in air and striking a plasma medium. The plasma has the properties of silver: $\omega_p = 2000$ THz, $\vartheta_c = 50$ THz. The propagating wave has a center frequency of 4000 THz. (**a,b**) show electric field in the computational domain at different times (**a**) = 0 fs, (**b**) = 17 fs, and (**c**) = 23 fs. (Left side with film thickness of 1000 cells, right side with film thickness of 2000 cells).

Figure 7 shows the electromagnetic energy density U as a function of the LBM grid cell number, ranging from 100 to 100,000 cells. Richardson's method [36] was used to compute the convergence error of our proposed method and the exact solution of the electromagnetic energy density is estimated by

$$E = \lim_{\delta x \to 0} E(\delta x) \approx \frac{2^n E\left(\frac{\delta x}{2}\right) - E(\delta x)}{2^n - 1} + L\left(\delta x^{n+1}\right) \tag{37}$$

with an error $L\left(\delta x^{n+1}\right)$ of order $n + 1$, where $\delta x = \frac{L}{N}$, as $L$ is the length of the computational domain, and N is the number of cells. Here, we analyze the errors in the order $n = 2$. Thus, the relative errors are computed as

$$L1 = \frac{1}{N} \sum_{i=1}^{N} \left| \frac{E(\delta x) - E}{E} \right| \tag{38}$$

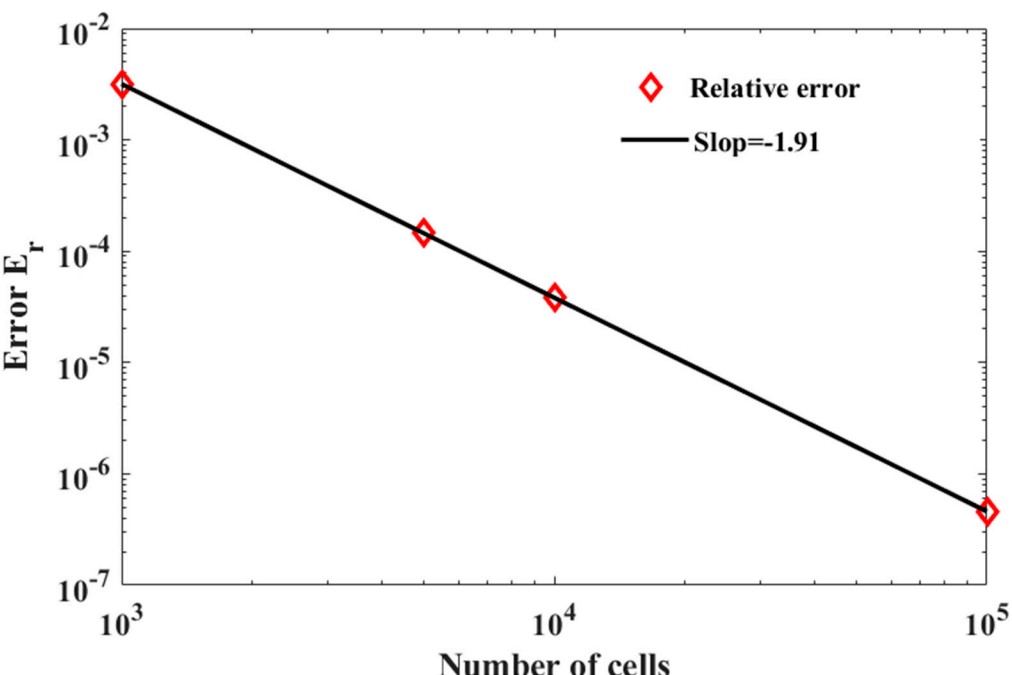

**Figure 7.** Numerical error as a function of the number of grid points.

Figure 7 shows that the relative error decrease with the increase of LBM grid cell number as $\delta x^{1.92}$. The decrease in errors verifies that the present scheme has a second-order convergence.

### 3.3. Effects of Layer Thickness on EM Waves in 1D Plasma PhCs

In this section, by using the proposed lattice Boltzmann scheme, we analyze the EM wave propagation behaviors in 1D plasma PhCs, demonstrate the adaptability of the proposed method, and present an interesting phenomenon of the plasma medium to incoming EM waves. By tuning the layer thickness $d$ of the 1D plasma PhCs, we can change the EM wave propagation behavior in 1D plasma PhCs.

Figure 8 shows a schematic of the proposed topological PhCs. 1D PhCs comprised alternating layers of plasma and air, with a plasma layer thickness of $d = 200$ nm (Figure 9), $d = 20$ nm (Figure 10), and $d = 2$ nm (Figure 11) in the region of $x = 0.6$ L to $x = 0.8$ L, L = 10 μm in this case.

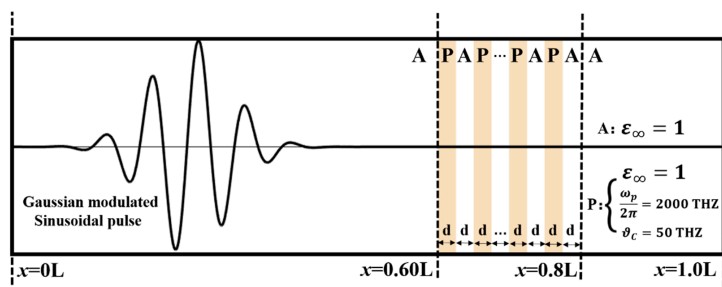

**Figure 8.** Schematic computational domain and incident Gaussian-modulated sinusoidal pulse through 1D plasma PhCs.

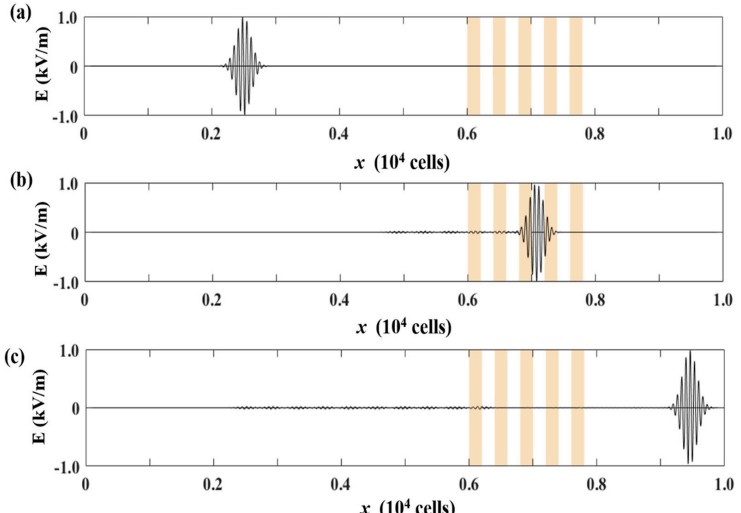

**Figure 9.** Simulation of a wave propagating in air and striking 1D plasma PhCs with plasma layer thickness $d = 200$ nm. The plasma has the properties of silver: $\omega_p = 2000$ THz, $\vartheta_c = 50$ THz. The propagating wave has a center frequency of 4000 THz. (**a,b**) show electric field in the computational domain at different times (**a**) = 0 fs, (**b**) = 15 fs, and (**c**) = 23 fs. (Left side with film thickness of 1000 cells, right side with film thickness of 2000 cells).

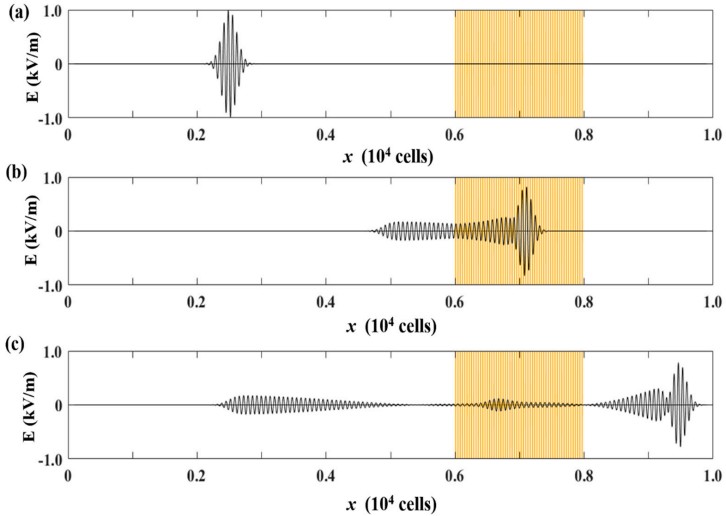

**Figure 10.** Simulation of a wave propagating in air and striking 1D plasma PhCs with plasma layer thickness $d = 20$ nm. The plasma has the properties of silver: $\omega_p = 2000$ THz, $\vartheta_c = 50$ THz. The propagating wave has a center frequency of 4000 THz. (**a,b**) show electric field in the computational domain at different times (**a**) = 0 fs, (**b**) = 15 fs, and (**c**) = 23 fs. (Left side with film thickness of 1000 cells, right side with film thickness of 2000 cells).

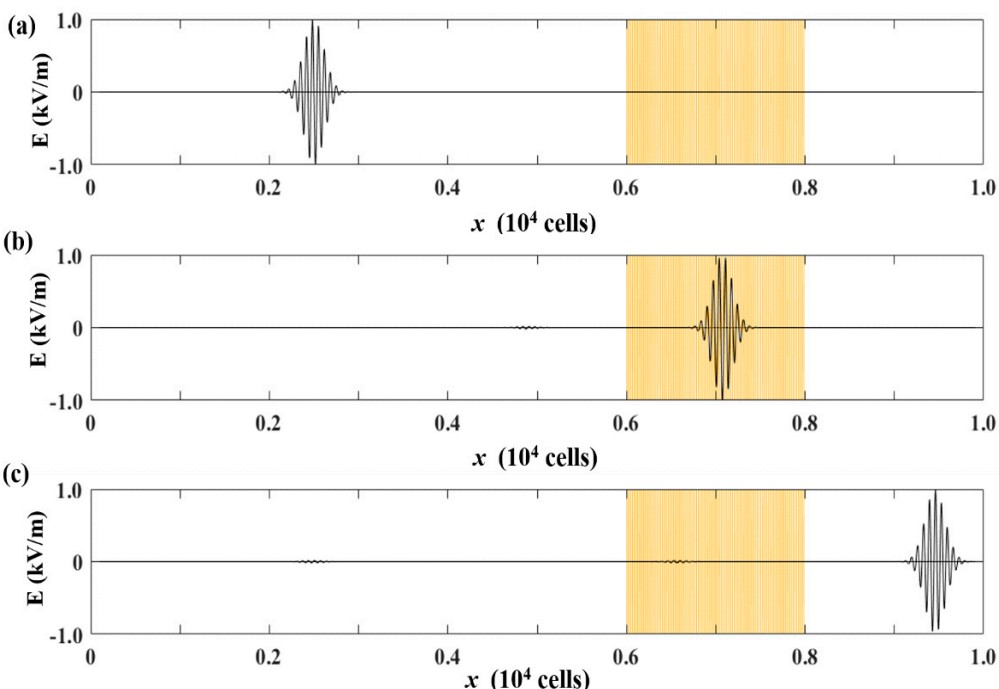

**Figure 11.** Simulation of a wave propagating in air and striking 1D plasma PhCs with plasma layer thickness $d = 2$ nm. The plasma has the properties of silver: $\omega_p = 2000$ THz, $\vartheta_c = 50$ THz. The propagating wave has a center frequency of 4000 THz. (**a,b**) show electric field in the computational domain at different times (**a**) = 0 fs, (**b**) = 15 fs, and (**c**) = 23 fs. (Left side with film thickness of 1000 cells, right side with film thickness of 2000 cells).

The simulation results for the thin, middle, and thick layer thicknesses are presented in Figures 9–11, respectively. Figure 9a shows that, at time $t = 0$ fs, the initial Gaussian-modulated sinusoidal pulse at a frequency of 4000 THz is in the air zone. At time $t = 15$ fs, Figure 9b shows that when the EM waves propagate through the two plasma layers on the left side, only a very small part of the incoming EM waves is reflected back to the air domain, and most of the other part penetrates the plasma; at time $t = 23$ fs, Figure 9c shows that the EM waves pass through the leftmost plasma layer and still move toward the right, and there are ten pulses with small amplitude propagating left. When the plasma layer thickness is decreased to $d = 20$ nm and $d = 2$ nm, the propagating EM waves in the computational domain at the same time snap as those in Figure 9 and are correspondingly shown in Figure 11. At t = 15 fs, Figure 10b shows that when the EM waves pass through the two plasma layers on the left side, a significant part of it is reflected back into the air with plasma thickness $d = 20$ nm, whereas when the EM waves propagate through the left side of the two plasma layers, only a very small part of the incoming EM waves is reflected back into the air domain with plasma layer thickness $d = 2$ nm as shown in Figure 11b. At $t = 23$ fs, Figure 10c shows that the EM waves are separated into two parts in the computational domain, one significantly larger part propagates to the left, and the other is located in the right air domain, which is the transmitted portion and still moves forward, while the EM waves almost completely pass through the plasma layers without an observable reflected part in the left air domain, as shown in Figure 11c. Since the center frequency $f_c$ of the incoming EM waves is 4000 THz, the center wavelength $\lambda_c = \frac{c}{f_c} = 75$ nm. The relative permittivity of plasma at 4000 THz is near that of vacuum, as shown in Figure 3. According to the theory of 1D plasma PhCs with constant permittivity, the most significant bandgap with $f_c = 4000$ THz is achieved when the plasma layer thickness $d \approx \frac{\lambda_c}{4} = 18.75$ nm. Figures 8–10 show that the proposed lattice Boltzmann scheme can be used to design the plasma layer thickness of 1D plasma PhCs.

## 4. Conclusions

In this study, a novel LBM-SEF method is introduced to simulate electromagnetic waves propagating in a 1D plasma medium. It is verified that the proposed method is mathematically consistent with Maxwell's equations by using the Chapman–Enskog expansion method. The accuracy of the simulation results from the proposed method is demonstrated through a comparison with the FDTD method. Two typical cases are executed to analyze the characteristics of electromagnetic waves that propagate through a plasma slab and 1D plasma PhCs. The results demonstrated the suitability of the proposed model for frequency-dependent reflection and transmission at the air–plasma interface. Moreover, it illustrated how to construct 2D and 3D LBM for electromagnetic waves in 1D plasma PhCs.

**Author Contributions:** Conceptualization, H.M. and M.T.; methodology, H.R.; software, W.J.; validation, W.G.; formal analysis, B.W. and Y.W. All authors have read and agreed to the published version of the manuscript.

**Funding:** This research was funded by the National Natural Science Foundation (grant numbers 42072163,12104513, and 21776315), Natural Science Foundation of Shandong Province (grant number ZR2019MD006), and Fundamental Research Funds for the Central Universities (grant number 19CX05001A).

**Acknowledgments:** The authors would like to thank the reviewers for their helpful comments and criticism.

**Conflicts of Interest:** The authors declare no conflict of interest.

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
