# Peer review of "A Novel Lattice Boltzmann Scheme with Single Extended Force Term for Electromagnetic Wave Propagating in One-Dimensional Plasma Medium"

_electronics, doi:10.3390/electronics11060882_

Round 1
Reviewer 1 Report
The authors have presented and demonstrated a numerical scheme with which to simulate a PDE for electromagnetic waves. I have a few suggestions:
(1) It would be very useful to quantify the dispersion error, e.g., along the lines of:
Numerical analysis of the lattice Boltzmann method for simulation of linear acoustic waves, Dhuri, D., Hanasoge, S. M., Perlekar, P., & Robertsson, J. O. A., Physical Review E, 95, 4, 043306
(2) The authors should cite previous papers that have attempted to simulate the electromagnetic wave equation using the Lattice Boltzmann method, such as, e.g., Lattice-Boltzmann Formulation for Electromagnetic Wave Propagation, Hanasoge, S. M., Succi, S. & Orszag, S. A., Europhysics Letters, 96, 14002 and other references.
(3) There are a number of minor errors
(a) Lines 200-201, the sentences ought to be linked (there's no verb in the opening sentence)
(b) Fig. 7 caption is wrong
(c) Medium is the singular form and media is the plural form. There are numerous instances where media is used incorrectly.
Author Response
Point 1: It would be very useful to quantify the dispersion error, e.g., along the lines of:
Numerical analysis of the lattice Boltzmann method for simulation of linear acoustic waves, Dhuri, D., Hanasoge, S. M., Perlekar, P., & Robertsson, J. O. A., Physical Review E, 95, 4, 043306
Response 1: Implemented. We have added the discussion about the dispersion relation using the mehtod proposed by Dhuri et al.,2017, please see line 278-295.
Point 2: The authors should cite previous papers that have attempted to simulate the electromagnetic wave equation using the Lattice Boltzmann method, such as, e.g., Lattice-Boltzmann Formulation for Electromagnetic Wave Propagation, Hanasoge, S. M., Succi, S. & Orszag, S. A., Europhysics Letters, 96, 14002 and other references.
Response 2: Implemented. We have cited previous papers that have attempted to simulate the electromagnetic wave equation using the Lattice Boltzmann method,please see line 70.
Point 3: There are a number of minor errors
(a) Lines 200-201, the sentences ought to be linked (there's no verb in the opening sentence)
(b) Fig. 7 caption is wrong
(c) Medium is the singular form and media is the plural form. There are numerous instances where media is used incorrectly.
Response 3: Implemented.
- We have corrected the sentence,please see line 249-250.
- We have corrected the caption , please see line 391.
- We have carefully checked the manuscript and corrected these places, furthermore, we also invited native english speaker to help us edit the language of this manuscript (The Elservier lanuage editing project for this manuscirpt is no.17085)
Reviewer 2 Report
The authors presented
- LBM should be defined when first time appears in the manuscript, either in abstract or introduction, instead in the material method.
- Could authors describe the limitation of the proposed model as well for the anisotropic plasma medium, it can be additional content in the discussion.
- The proposed model seems in very good agreement with the FDTD model. In what condition the LBM will be more relevant or predict more accuracy for the numerical study compared to the FDTD? please discuss it in more detail.
Author Response
Point 1: LBM should be defined when first time appears in the manuscript, either in abstract or introduction, instead in the material method.
Response 1: Implemented. We have defined LBM in introduction (Please see line 61)
Point 2: Could authors describe the limitation of the proposed model as well for the anisotropic plasma medium, it can be additional content in the discussion..
Response 2: Implemented. We have added more discussion about the limitation of the proposed model in the manuscript (Please see line 235-240 and line 278-291).
Point 3: The proposed model seems in very good agreement with the FDTD model. In what condition the LBM will be more relevant or predict more accuracy for the numerical study compared to the FDTD? please discuss it in more detail.
Response 3: Implemented. We have added more discussion about the dispersion error of FDTD model and the LBM model. (Please see line 284-293).
Reviewer 3 Report
Title: A Novel Lattice Boltzmann Scheme with Single Extended Force Term for Electromagnetic Wave Propagating in One Dimensional Plasma Medium MS NO: electronics-1601355
In this study, a novel LBM with a single extended force term is introduced to simulate the electromagnetic waves propagating in 1D plasma medium. The work and the presented results are novel, applied, and are of great importance to the related audience. I strongly recommend it to be accepted for publication in this journal after addressing the following comments.
· Explain more about the advantages of the LBM over finite difference time domain method for this particular problem. · Discuss further the limitations of the proposed method. · In section 1, line 63, the application of LBM is listed. Simulation of heat transfer phenomena should also be added for readers' information. The followings are some examples: DOI: 10.22055/JACM.2019.29503.1605; 10.1016/j.camwa.2021.01.016 · Did you do any mesh grid study for numerical simulation? · Immersed boundary - lattice Boltzmann method is another advanced method that can be used for simulation of waves or particles in different mediums. It is suggested to introduce this technique in the introduction. Referring following papers will be useful: DOI: 10.1007/s10973-019-08329-y; 10.1016/j.molliq.2020.114941
|
Author Response
Point 1: In this study, a novel LBM with a single extended force term is introduced to simulate the electromagnetic waves propagating in 1D plasma medium. The work and the presented results are novel, applied, and are of great importance to the related audience. I strongly recommend it to be accepted for publication in this journal after addressing the following comments.
Response 1: Thank you very much.
Point 2: Explain more about the advantages of the LBM over finite difference time domain method for this particular problem.Discuss further the limitations of the proposed method.
.Response 2: Implemented. We have added more discussion about the limitation of the proposed model , and the advantages of the LBM over finite difference time domain in the manuscript (Please see line 235-240 and line 278-293).
Point 3: In section 1, line 63, the application of LBM is listed. Simulation of heat transfer phenomena should also be added for readers' information. The followings are some examples:DOI: 10.22055/JACM.2019.29503.1605; 10.1016/j.camwa.2021.01.016.
.Response 3: Implemented. We have cited the references about LBM based heat transfer simulation.(Please see line 64)
Point 4: Did you do any mesh grid study for numerical simulation?.
.Response 4: Implemented. We use equally divided lattice cell grid type in this study, and have analyzed the effect of cell number on simulation erros (Please see line 365-376)
Point 5: Immersed boundary - lattice Boltzmann method is another advanced method that can be used for simulation of waves or particles in different mediums. It is suggested to introduce this technique in the introduction. Referring following papers will be useful: DOI: 10.1007/s10973-019-08329-y; 10.1016/j.molliq.2020.114941
..Response 5: Implemented. We have introduced the immered boundary-lattice Boltzmann mthod in the introduction. (Please see line 64-67)
Reviewer 4 Report
This article develops a new Lattice Boltzmann Method for simulating the electromagnetic wave propagation in 1D plasma media which is based on what the authors call “a single extended force term”. The consistency of the model is confirmed conducting a Chapman-Enskog analysis (CEA). The model is successfully validated using different representative benchmark scenarios.
The topic investigated in this article is of interest to researchers with theoretical background. Although, the method, the results, and most of the figures are helpful, the manuscript is not well structured and is unclearly written. Many typos occur and the formatting of equations and variables as well as abbreviations used in the text are not consistent and need extensive editing. Moreover, this article shows many similarities (regarding concept, CEA and simulation setups) with Ref. [16].
All in all, I cannot recommend the publication of the article in its current form. Instead I recommend that the authors double check formatting, typos and equations, as this should not be the task of a reviewer. It just distracts the reader from the content.
More specific remarks and small comments are given below:
GENERAL COMMENTS:
- For the simulations, the relaxation time \tau and the weighting factors A and B are not specified. Thus, it is not possible to reproduce the results.
- The unit conversion (from LBM units to SI units) should be summarized in a table. Otherwise it is totally unclear how the units (kV/m), (s) and (Hz) can be determined from the simulations.
- Many careless mistakes and typos occur in the text. Please double check again!
SMALL REMARKS:
- Abstract:
- lattice Boltzmann instead of Lattice Boltzmann
- Chapman-Enskog instead of Champman-Enskog
- Once abbreviations have been introduced, please use them consistently in the following parts of the text: e.g. 1D in line 48
- In line 64: The abbreviation LBM has not yet been introduced; it is instead introduced in line 81
- (3): wrong sign, either source-free (without J) or current density J is missing, although mentioned in line 101
- In line 109 & 112: Chen et al. instead of Chen. et al.
- In line 128: e_i is the lattice velocity
- (10): strange typesetting; index “i” missing for A_i and B_i
- (11): Eq. (11) is a duplication of Eq. (8)
- Line 141: consistent nomenclature Eq. (10) and Eq. (11) instead of Equ. (10) and Equ. (11)
- (12): index “i” missing for B_i
- (14): consistent nomenclature f^0 instead of f^eq
- (16) & Eq. (18) & Eq. (20): consistent nomenclature O(\Delta t^3) instead of o(\Delta t^3)
- (17): redundant information in 2nd line
- (19): inconsistent nomenclature
- (26): 2A\eps\eps_0 = 1 / \mu instead of 2A\eps\eps_0 = \mu
- The subsection 2.5 is too short and not very self-explanatory
- In line 229ff.: unit Hertz is Hz instead of HZ
- In line 231: word missing at the end, totally incomprehensible sentences
- Figure 3 and Figure 7: damping frequency \theta_c = 57 THz, while in the text it is \theta_c = 50 THz
- (34): v_c is not explained, did you mean \theta_c instead?
Author Response
Point 1: For the simulations, the relaxation time \tau and the weighting factors A and B are not specified. Thus, it is not possible to reproduce the results..
Response 1: Implemented. We have specified the the relaxation time \tau and the weighting factors A and B , please see line 138-140 and line 211-215.
Point 2: The unit conversion (from LBM units to SI units) should be summarized in a table. Otherwise it is totally unclear how the units (kV/m), (s) and (Hz) can be determined from the simulations.
Response 2: Implemented. We have added the unit conversion table (table 1), please see line 242-247.
Point 3: Many careless mistakes and typos occur in the text. Please double check again.
Response 3: Implemented. We have carefully checked the manuscript and corrected these places, furthermore, we also invited native english speaker to help us edit the language of this manuscript (The Elservier lanuage editing project for this manuscirpt is no.17085)
Point 4: Abstract: lattice Boltzmann instead of Lattice Boltzmann ,Chapman-Enskog instead of Champman-Enskog.
Response 4: Implemented. We corrected these places, please see line 16-33.
Point 5: Once abbreviations have been introduced, please use them consistently in the following parts of the text: e.g. 1D in line 48.
Response 5: Implemented. We have corrected these places and used abbreviations in the following parts of the text, including 1D, LBM, LBM-SEF and PhCs.
Point 6: In line 64: The abbreviation LBM has not yet been introduced; it is instead introduced in line 81.
Response 6: Implemented. We have corrected this place, and introduced LBM in 61, please see line 61.
Point 7: wrong sign, either source-free (without J) or current density J is missing, although mentioned in line 101
Response 7: Implemented. We use source-free form and corrected the sentence, please see line 102-109.
Point 8: In line 109 & 112: Chen et al. instead of Chen. et al.
Response 8: Implemented. We corrected these two places, please see line 116 and 120.
Point 9: In line 128: e_i is the lattice velocity.
Response 9: Implemtend. We have corrected this place to lattice velocity, please line 136-137.
Point 10: Eq.(10) strange typesetting; index “i” missing for A_i and B_i.
Response 10: Implemented. We have add a sentence to introduce A and B, please see line 148-149.
Point 11: Eq. (11) is a duplication of Eq. (8)
Response 11: Implemented. We have deleted one.
Point 12: consistent nomenclature Eq. (10) and Eq. (11) instead of Equ. (10) and Equ. (11)
Response 12: Implemented. We have corrected this sentence,please see line 154-155.
Point 13: (12): index “i” missing for B_i
Response 13: Implemented. We have corrected this term, please see line 155.
Point 14: (14): consistent nomenclature f^0 instead of f^eq
Response 14: Implemented. We have corrected this place, please see line 168.
Point 15: (16) & Eq. (18) & Eq. (20): consistent nomenclature O(\Delta t^3) instead of o(\Delta t^3)
Response 15: Implemented. We have have corrected thsese places, please see line 175-190.
Point 16: (17): redundant information in 2nd line
Response 16: Implemented. We have delted the redundant infromation, please see line 178.
Point 17: (19): inconsistent nomenclature
Response 17: Implemented. We have corrected this place to keep inconsistent nomenclature, please see line 184.
Point 18: (26): 2A\eps\eps_0 = 1 / \mu instead of 2A\eps\eps_0 = \mu
Response 18: Implemented. We have corrected this place, pleas see line 208.
Point 19: The subsection 2.5 is too short and not very self-explanatory
Response 19: Implemented. We have combined section 2.5 to section 2.4, please see line 226-247.
Point 20: In line 229ff.: unit Hertz is Hz instead of HZ
Response 20: Implemented. We have corrected this place, please see line 299.
Point 21: In line 231: word missing at the end, totally incomprehensible sentences
Response 21: Implemented. We have corrected this sentence, please see line 301-302.
Point 22: Figure 3 and Figure 7: damping frequency \theta_c = 57 THz, while in the text it is \theta_c = 50 THz
Response 22: Implemented. We have corrected the damping frequency in Figure 3 and Figure 7, please see line 322 and 391.
Point 23: (34): v_c is not explained, did you mean \theta_c instead?
Response 23: Implemented. We have added the explaination of this parameter, please see line 314-316.
Round 2
Reviewer 3 Report
All the comments are addressed.
Author Response
Point 1: All the comments are addressed.
Response 1: Thank you very much.
Reviewer 4 Report
Although, the method, the results, and most of the figures are helpful, the manuscript is still not well structured and is unclearly written. Most of the typos and formatting errors have been corrected. However, the article shows still many similarities with Ref. [16]. The Chapman-Enskog analysis, parametrization of the model and structure of the paper was simply taken from Ref. [16]. Thus, the novelty of the method is questionable and the scientific contribution of the authors remains unclear.
I am not very convinced about the novelty and significance of the content.
Two remarks remain:
- Table 1: consistency ‘py’ vs.’Py’
- Line 136: e_i is the lattice velocity vector, not the lattice vector. See Eq. (7): it has velocity units
Author Response
Point 1: Although, the method, the results, and most of the figures are helpful, the manuscript is still not well structured and is unclearly written. Most of the typos and formatting errors have been corrected. However, the article shows still many similarities with Ref. [16]. The Chapman-Enskog analysis, parametrization of the model and structure of the paper was simply taken from Ref. [16]. Thus, the novelty of the method is questionable and the scientific contribution of the authors remains unclear.
Response 1: Implemented. Thanks for your comments. In the studies by Chen et al. [16], it is necessary to calculate the pseudo permittivity in advance before simulating the terahertz electromagnetic wave propagation, while the accuracy of the pseudo permittivity is related to the simulation time step ∆t used in special cases [16]. Based on Eq. (5), it is no longer necessary to use pseudo permittivity. It can be observed from the following mathematical derivation process that only one force term is needed, without the use of pseudo permittivity. Furthermore, we estabished a new LBM scheme with only single force term which is not depened on the pseudo permittivity proposed by Chen et al. [16]. Besides the above, we also investigated the EM waves propagation in 1D plasma PhCs with the new proposed LBM-SEF mthod in this study. Please see line 72-76, 120-125, 380-440 .
Point 2: Table 1: consistency ‘py’ vs.’Py’
Response 2: Implemented. We have addressed these places, please see line 247-248.
Point 3: Line 136: e_i is the lattice velocity vector, not the lattice vector. See Eq. (7): it has velocity units
Response 3: Implemented. We have addressed this place, please see line 137.